# Adoptability of digital payments for community health workers in peri-urban Uganda: A case study of Wakiso district

Veronica Kembabazi[1]*, Arnold Tigaiza[1], Charles Opio[1], Juliet Aweko[1], Mary Nakafeero[2], Fredrick Edward Makumbi[3], Michael Ediau[1], Elizabeth Ekirapa Kiracho[1], Andrew K. Tusubira[4,5], Peter Waiswa[2,6,7]

1 Department of Health Policy, Planning and Management, Makerere University School of Public Health, Kampala, Uganda, 2 Makerere University School of Public Health, Kampala, Uganda, 3 Department of Epidemiology and Biostatistics, Makerere University School of Public Health, Kampala, Uganda, 4 Department of Community Health and Behavioral Sciences, Makerere University School of Public Health, Kampala, Uganda, 5 The Medical Research Council/Uganda Virus Research Institute (MRC) & London School of Hygiene and Tropical Medicine (LSHTM) Uganda Research Unit, Entebbe, Uganda, 6 Global Public Health, Karolinska Institute, Stockholm, Sweden, 7 Busoga Health Forum, Jinja, Uganda

* kdalene3@gmail.com

**Data Availability Statement:** All relevant data are within the manuscript and its Supporting information files.

## Abstract

### Background

Whereas digital payments have been identified as a solution to health payment challenges, evidence on their adoptability among Community Health Workers (CHWs) is limited. Understanding their adoptability is crucial for sustainability. This study assessed the adoptability of digital payments for CHWs in Wakiso district, Uganda.

### Methods

A convergent parallel mixed-methods study was conducted between November and December 2022, in Wakiso district, Uganda. We surveyed a random sample of 150 CHWs using a structured questionnaire and conducted key informant interviews among three purposively selected Digital payment coordinators. The study utilized the Technology Acceptance Model (TAM) framework to assess the adoptability of digital payments among CHWs. Factor analysis was performed to extract composite variables from the original constituting variables. Using the median, the outcome was converted to a binary variable and logistic regression was conducted to assess the association between the TAM constructs and adoptability of digital payments by CHWs. Quantitative data was analyzed using STATA 14, while qualitative data was transcribed verbatim and analyzed using ATLAS.ti 22.

### Results

Nearly all participants (98.0%; n = 49) had previously received payments through mobile money, a digital payment method. (52%; n = 78) of CHWs said they intend to use digital payment modalities. Perceived risk of digital payments was associated with 83% lower odds of adoptability of digital payment modalities (OR = 0.17;95%CI:0.052, 0.54), while perceived

**Funding:** Financial support for this study was obtained through the Digital Health Payments Initiative and Research project (DHPI-R) at Makerere University funded by the Bill&Melinda Gates Foundation (grant number: INV-03047). The funders had no role in study design, data collection and analysis, decision to publish, or preparation of the manuscript.

**Competing interests:** The authors have declared that no competing interests exist.

trust had nearly three times higher odds of adoptability of digital payment modalities (OR = 2.82;95%CI:1.41, 5.67). Qualitative interviews showed that most CHWs reported positive experiences with digital health payments, including effectiveness and completeness of payments except for delays associated with mobile money payments across payment providers. Mobile money was reported to be easy to use, in addition to fostering financial responsibility compared to cash.

## Conclusion

CHWs in Wakiso district intend to use digital payment modalities, particularly mobile money/e-cash. Perceived risk of the payment method and trust are key determinants of adoptability. Synergized efforts by both payment providers to manage payment delays and mitigate risks associated with digital payments could attenuate perceived risk and build trust in digital payment modalities.

## Introduction

Digital payment, also known as electronic payment, involves the transfer of money or digital currency from one account to another using various digital payment technologies. These technologies include mobile payment apps like PayPal, as well as other electronic payment systems such as bank transfers, debit cards, and electronic(e)-money such as mobile money [1]. Mobile money refers to financial transactions and services that can be carried out using a mobile device particularly a mobile phone or tablet [2]. Another widely adopted technology in the realm of digital payment is Unstructured Supplementary Service Data (USSD), which is a protocol utilized within the Global System for Mobile Communications (GSM) to initiate actions like money withdrawals, payment of bills, and purchases using electronic money through text messages. Mobile money (MM)/e-cash technology has become prevalent in developing nations like Uganda, particularly among Telecom Service Providers (TSPs). Traditional Financial Institutions such as banks have also embraced this trend by developing mobile applications to offer customers a wide range of services remotely [3].

The unprecedented consequences of the COVID-19 pandemic, which involved movement restrictions and the need for contactless money transfers, further accelerated the demand and acceptance of digital payments, serving as a clear indication of their potential for widespread adoption [4, 5]. There is a growing trend led by global agencies such as the World Health Organization (WHO), the vaccine alliance GAVI and the Global Fund, to move towards digital payments in the health sector in low- and middle-income countries (LMICs) [6]. This move aims at digitizing payments for health workers involved in vaccination and immunization campaigns [6] where Community Health Workers (CHWs) are a critical group. CHWs have been acknowledged globally as key mobilizers of their communities for vaccination according to the WHO Polio Eradication Strategy 2026 [7]. In Uganda, the use of digital payments for Community Health Workers has been mainly project-oriented [8] although the Ministry of Health sometimes uses digital payments for a few health campaigns conducted in the country using the Electronic Receipting and Invoicing System (EFRIS) [9] and MM for CHWs.

Transition to digital payments not only addresses the challenges associated with cash-based modalities for frontline health workers [10] but also brings several additional benefits including transparency and security of funds and easy traceability and accountability of financial

transactions [8]. Digital payment also contributes to cost savings, promotes financial inclusion, and facilitates the economic participation of women [11, 12]. By creating an auditable transactional record, digital payments have the potential to reduce fund leakage, eliminate the costs of security and storage, and improve overall health system management [12–15]. Indeed, the prompt digital payment of CHWs during the COVID-19 response and Polio vaccination campaigns in Côte d'Ivoire and Uganda demonstrated the effectiveness of digital payments in enhancing field duties and overall performance [6, 10]. CHWs in many countries are, however, not part of the salaried workforce and are still paid largely through cash for community health activities that happen occasionally. Therefore, the adoption of digital payments for CHWs is timely, given the availability of mobile money transfers run by telecommunication companies, with connectivity across the country and only requiring one to have access to a phone. It is also important, to understand the mechanisms that will underscore the adoptability of these payment methods/modalities in low-resource settings.

Despite the benefits of using digital payment modalities, several barriers to the utilization of these payment modalities exist and need to be addressed. One significant challenge is the lack of financial systems such as bank accounts, credit cards, and mobile payment apps, particularly among individuals residing in rural areas of developing countries [16]. However, the emergence of bank-independent e-money payment options, such as mobile money, offers promising opportunities for more health workers to access digital payments. This shift towards digital payments presents an opportunity to overcome some of these challenges and enhance the effectiveness and efficiency of community health activities such as health campaigns [6, 8, 9]. Literature, nonetheless, indicates that digital payments have not been widely adopted in the healthcare sector for the CHWs in LMICs [6, 17].

Additionally, the varying levels of education among CHWs raise concerns about their ability to comprehend and effectively use these digital payment platforms. The intention to embrace digital payment methods among such a cadre in Uganda's healthcare system today remains an abstract question. In Wakiso, a district encircling the capital city of Uganda, the possibilities for access and utilization of digital payments modalities are higher than in rural areas [5]. But the above-mentioned challenges, make it crucial to determine whether digital payments are an appropriate payment modality for CHWs in Wakiso district. Besides, the per-urban nature of Wakiso district gives a picture of both the urban and not-so urban perspectives in response to the research question of adoptability. It is also necessary to identify the factors that can be addressed to improve the adoption and application of digital payments in various settings in Uganda. This study aimed at assessing the prospective adoptability of digital payments among CHWs in Wakiso and exploring their experiences of using digital payment modalities.

## Materials and methods

### Study site

This study was conducted in Wakiso district which surrounds Kampala, Uganda's capital, and covers about 1,907 km$^2$. Wakiso district, which hosts 46% of all formal employment in Uganda was selected because it is a densely populated peri-urban area—now transitioning to an urban area—which contributes significantly to Uganda's overall Gross Domestic Product (GDP) [18] and would provide a fair picture of the ability of urban based community health workers to adapt to digital payments.

### Study population

This study was conducted among CHWs carrying out community health activities in Wakiso district supported by various Implementing Organizations including Infectious Diseases

Institute (IDI) Central and Uganda National Expanded Program on Immunization (UNEPI), PACE, AMREF and Living Goods, and being paid by these organizations through digital payment systems.

A Community Health Worker is a member of a community, chosen by community members or organizations and capable of providing preventive, promotional, and rehabilitation care to that community. In Uganda, the CHW is a member of a Village Health Team (VHT). In-depth interviews were conducted with CHWs with varied experiences with digital payments and willing to share more information on the subject. Key informant interviewees were coordinators of CHWs and individuals who play key roles in the management of digital payments for CHWs such as payment providers. Coordinators of these CHWs (facility health assistants) were interviewed to provide information as Key Informants on the experiences, usefulness and ease of use and risk of paying CHWS using digital payments compared with cash-based payments. The study included CHWs currently volunteering at the district and excluded those who were scheduled to retire within a year from the time of the interview since they would no longer receive digital payments after a year and the study findings would not affect them.

## Study design

A convergent parallel mixed-methods study was conducted using an adapted version of the Technology Acceptance Model (TAM) [19] and the Unified Theory of Acceptance and Use of Technology (UTUAT) [20] (Fig 1) to assess the adoptability of digital payments among CHWs in Wakiso district. Adoptability of digital payments refers to the prospective intention of the CHWs to continue using the digital payment method and the determining factors.

The TAM is a well-known model, created by Fred Davis in 1986, that has been used in numerous studies around the globe [19, 21–26] to determine behavioral intention to use technology and online payment systems. It is useful for identifying the modifications which must be made to a digital system to make it acceptable/adoptable to users [27]. The model was selected to fit the local context since its four key factors; perceived usefulness, perceived risk, perceived ease of use and trust may independently have a relationship with the adoptability of a digital payment modality in Uganda and may also have a relationship with other factors in the TAM depending on the population and the system to be adopted by users. Adoptability is considered to have a decisive effect on user behavior and can therefore guide studies on the application of e-services.

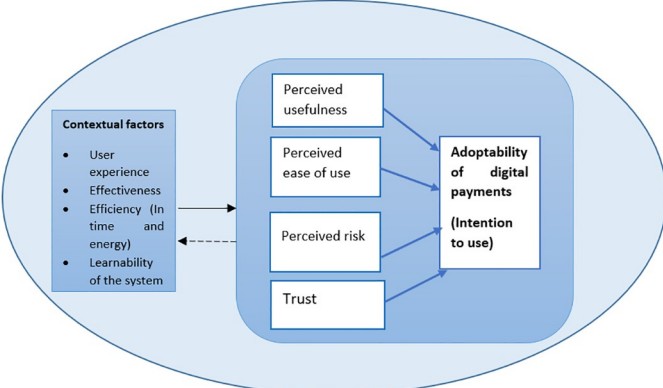

**Fig 1. Factors influencing adoptability of digital payments.** Adapted from Technology Acceptance Model by Abd Malik and Syed Annuar [19,39].

## Sample size determination

The survey sample size was calculated using the Kish Leslie formula [28]. The estimated conservative prevalence of adoptability of DHPs was assumed to be 50% due to a lack of evidence on the prevalence of adoptability of DHPs. A z-score of 1.96 corresponding to the 95% confidence interval, a precision of 8%, and a non-response rate of 10% was used to obtain a sample size of 150 community health workers. For the In-depth and KI interviews, 11 and three [3] interviews were conducted respectively. Approximately two [2] CHWs were interviewed in each of the six randomly selected sub-counties of the district. In-depth interviewees included CHWs, while Key Informant interviewees comprised individuals based at an organization that funded CHWs and paid CHWs digitally.

## Sampling procedure

Simple random sampling was conducted using a line list of community health workers who were previously paid using digital payments for vaccination and other health campaigns under two [2] NGOs actively operating in the study area. A line list of Community Health Workers was obtained according to previous use of digital payments. The maximum variation sampling strategy was employed using sub-county and parish worker designations to select the participants for the IDIs and KIIs until saturation was achieved.

## Study variables

The study variables included adoptability of digital payments (measured as intention to use digital payments), perceived usefulness (PU), perceived ease of use (PEU), perceived risk (PR) and trust (T), and context. The variables in the model for this study were adopted particularly from Abd Malik and Syed Annuar [19] but are similar to other studies on the intention to use digital payment modalities. Additional data on age, sex and education level of the CHWs was collected.

**Adoptability.** Users will more certainly adopt a digital method, if they are more intent on using it. Tomić, Kalinić [29]. This study aimed to assess the adoptability of digital payments for CHWs in Wakiso in order to determine the prospective intention of these CHWs to continue using the digital payment methods.

**Perceived usefulness (PU).** PU can be defined as users' perception of how a system (digital payment) enhances their productivity and performance. Many authors hold the view that the perceived usefulness of a system has a positive influence on the user's intention to use it and eventually its usage [30], therefore it is usually the most influential factor in adoption [24, 31].

**Perceived ease of use.** According to Davis, Bagozzi [32], PEU is the degree of ease associated with the use of the system. This construct was found to be a significant predictor of intention to use digital systems in some studies, and not in others [30].

**Perceived risk.** It is defined as the user's perception of the uncertainty and the adverse consequences of using digital payments [19]. A study that used Perceived Technology security, which refers to secure operation, payment, and safety (similar to Perceived risk) found it to be the most important variable influencing the intention to use cashless payments/modalities.

**Trust.** The findings of Abd Malik and Syed Annuar [19], Mensah [30] and Tomić, Kalinić [29] who use the TAM and UTUAT models, respectively, both show that Trust is a significant factor influencing intention to use digital services. Pavlou [33] and Tomić, Kalinić [29] refer to trust as an important element in uncertain environments such as e-commerce and consider it very subjective.

## Data collection and management

Between 30[th] November and 6[th] December 2022, data on user experiences, perceived usefulness, perceived ease of use and perceived risk of digital payments was collected from CHWs and payment providers. Prior to collecting the data, the investigators pretested and reviewed the data collection tools to ensure internal validity. Experienced Research Assistants (RAs) fluent in English and the local language (Luganda), and trained in the study procedures screened participants for eligibility and conducted both quantitative and qualitative data collection interviews. After selection of participants for the interviews with assistance of the District Health Office, appointments were scheduled with the Health Assistants and Community health workers, and then interviews were conducted in a convenient location. The interviews lasted about 20 minutes and were administered using tools translated in the language most convenient for the participant. The quantitative data was collected using the Kobo Collect mobile application the key informant and in-depth interviews were also recorded.

As part of quality control, the filled tools were reviewed for completeness and errors found were corrected to enhance the validity of the data collected.

The cleaned data was imported into STATA 15 for statistical analysis and data quality checks were conducted, including verification of all values to make sure they were permissible and to identify outliers. For value verification, the data was examined to identify values that fall outside the predefined cut-off values that represent real value sets. Qualitative data from the IDIs and KIIs was transcribed verbatim by experienced research assistants and entered in ATLAS.ti software. The audio recordings from the KIIs were stored on an external storage device for safety and ethical purposes.

## Data analysis

Quantitative data was analyzed using STATA 15. Descriptive statistics including means and standard deviations were used to summarize continuous variables.

**Measurement of constructs and exploratory results.** PU, PEU, PR, Trust and the outcome variable adoptability were then treated as composite variables, each being measured by a set of items. The questions were generated based on a review of literature on adoption of digital systems where the TAM and UTUAT model [16] were used. Each of the questions under the different constructs/variables, were assessed using a 5-point likert scale from Strongly agree to Strongly disagree to allow participants to express their views on each of the questions (items) (Table 1).

Using the raw average inter-item correlation and Cronbach's alpha, the internal consistency reliability of the scales was assessed considering that raw internal consistency measurement may give better values [34, 35]. Average Inter-item consistency values were all between 0.15 and 0.50 as suggested by Watson [35]. Cronbach's values were all above 0.5, and considered acceptable VIF values were all less than 5, hence collinearity was ruled out. Discriminant validity was assessed by calculating the correlation—Variance Inflation Factor (VIF)—among the constructs (Table 2).

**Logistic regression and hypothesis testing.** Summary analyses were conducted for the composite outcome variable, adoptability, with a minimum value of -4.159927 and a maximum value of 0.6140383. Using the median value (0.3134086), a new binary variable was developed from the outcome with the values below the median defined as "not intending to use" (48%), while those at the median and above as "intending to use" (52%) digital payments.

A model was developed using logistic regression to assess the influence of the various constructs on the intention to use digital payments and to assess the relationship between the

**Table 1. Reliability analysis.**

| Variable and sources | Items | FL | AIC | Alpha |
|---|---|---|---|---|
| Perceived Usefulness | [PU1] With digital payment systems, I always receive my money for the health activities | 0.606 | 0.28 | 0.67 |
| | [PU2] Digital payments enable me to receive payment on time | 0.601 | | |
| | [PU3] Overall, digital payments would support me in coping with work challenges/problems | 0.507 | | |
| | [PU4] Digital payments increase my performance at work and I get the expected results | 0.476 | | |
| | [PU5] Digital payments enable me to conduct tasks such as money transfers, and shopping, more easily | 0.460 | | |
| Perceived Ease of Use | [PEU1] Using digital payments is easier than cash-based payment because it allows me to access my payment faster | 0.333 | 0.46 | 0.72 |
| | [PEU2] I can easily navigate the digital system to access my electronic money | 0.862 | | |
| | [PEU3] Digital payment systems are clear and easily comprehensible to me | 0.857 | | |
| Perceived Risk | [PR1] Digital payments are reliable compared to cash-based payments | 0.604 | 0.23 | 0.54 |
| | [PR2] I am sure of getting my payment when paid using digital systems | 0.473 | | |
| | [PR4] I am concerned about the privacy of my personal information when using my payments systems | -0.164 | | |
| | [PR5] I feel that I am in control of my money when paid digitally | 0.614 | | |
| Trust | [T1] Digital payments service providers like Mobile money operators are credible and safe | 0.446 | 0.24 | 0.49 |
| | [T2] Digital payments network providers like MTN and Airtel are widely acknowledged and can be trusted | 0.575 | | |
| | [T3] I trust the digital payment providers (e.g., IDI, UNEPI) | 0.365 | | |
| | [T5] Digital payments systems are reliable in my area of work | 0.436 | | |
| Adoptability | [A2] I can use digital payment systems regularly | 0.532 | 0.40 | 0.77 |
| | [A3] Using digital payment systems is good for me | 0.710 | | |
| | [A4] Modern life involves the use of digital payment systems | 0.457 | | |
| | [A5] I would recommend digital payment systems to others | 0.703 | | |
| | [A7] I am willing/okay with being paid through digital systems by my employer in the future | 0.720 | | |

Note: FL (Factor Loading), AIC (Average Inter-item Correlation), Alpha (Cronbach's alpha)

various constructs in the model. All independent variables whose p-value was less than 0.25 were included in the model; all the p-values of the independent variables were less than 0.001.

For the qualitative component, thematic analysis, employing mainly a deductive process was used in analyzing the data and ATLAS.ti software was used to ease the analysis. The coding process was done by the two [2] lead researchers. The initial codes were generated based on the adapted Technology Acceptance Model constructs. Thereafter the transcripts were reviewed and emerging codes were identified and discussed, and the codebook was refined. The transcripts were then coded, coding was reviewed and codes were arranged and merged to develop themes and sub-themes. The coded themes and sub-themes were then exported to Microsoft Word for interpretation of data.

**Table 2. Discriminant validity.**

| | PU | PEU | PR | TRUST |
|---|---|---|---|---|
| PU | 1.000 | | | |
| PEU | 0.4220 | 1.000 | | |
| PR | -0.689 | -0.4731 | 1.000 | |
| TRUST | 0.291 | 0.1395 | -0.365 | 1.000 |

Factor analysis was performed to extract composite variables from the original constituting variables where only one factor was retained for each construct. Kaiser-Meyer-Olkin (KMO) statistics were assessed for each construct to determine appropriateness for data reduction.

### Ethical considerations

Ethical clearance was obtained from the Makerere University School of Public Health Higher Degrees Research and Ethics Committee. Administrative clearance was obtained from Wakiso district health office. Written informed consent was sought from each participant before the interview.

## Results

The study surveyed 150 respondents: the majority 69.4% (104) were 40 years and above; 82% (123) had secondary level education, and over 88% (132) were female. Nearly all participants, 98.0% (147) had ever received payments through mobile money (Table 3).

The regression results indicated a good model of fit: the measurement model was statistically significant with a goodness-of-fit p-value above 0.05, and 80% of the observations were correctly predicted in the intention to use digital payments.

Perceived Risk and Trust had a significant influence on the intention to use digital payments. CHWs who perceived digital payments as risky were 83% less likely to intend to use digital payment modalities (OR = 0.167, $p$ <0.01, 95% CI: 0.052–0.537), while CHWs who trusted digital payments were more than twice as likely to intend to use digital payment modalities (OR = 2.823, $p$ < 0.01, 95% CI: 1.406–5.670). PU and PEU were not associated with the adoptability of digital payment modalities among CHWs in Wakiso district (Table 4).

From the qualitative findings, we also generated sub-themes under the main constructs and these are summarized below. (Table 5).

**Table 3. Characteristics of the community health workers and their use of digital health payments.**

| Variable | Category | Frequency (n = 150) | Percentage (%) |
|---|---|---|---|
| Age | 20–29 | 11 | 7.3 |
| | 30–39 | 35 | 23.3 |
| | 40–49 | 52 | 34.7 |
| | 50+ | 52 | 34.7 |
| Education level | Primary | 18 | 12.0 |
| | Secondary | 123 | 82.0 |
| | Tertiary | 9 | 6.0 |
| Gender | Male | 18 | 12.0 |
| | Female | 132 | 88.0 |
| Method of learning to use payment systems | Self-taught | 92 | 61.3 |
| | Colleague | 15 | 10.0 |
| | Family member | 28 | 10.0 |
| | Employer | 15 | 18.7 |
| The system through which payments are received | Bank transfer | 1 | 0.7 |
| | Mobile money | 147 | 98.0 |
| | Other | 2 | 1.33 |
| Name of the payment provider | IDI | 46 | 30.7 |
| | UNEPI (MOH) | 18 | 12.0 |
| | Others* | 86 | 57.3 |

Others

*: PACE, Living Goods, ICCM

**Table 4. Factors determining adoptability of digital payments for payment of community health workers in Wakiso district.**

| Factor | Unadjusted | | | Adjusted | | |
|---|---|---|---|---|---|---|
| | OR | 95% CI | p-value | OR | 95% CI | p-value |
| Perceived usefulness | 3.67 | 2.14–6.27 | 0.000 | 1.09 | 0.52–2.28 | 0.828 |
| Perceived ease of use | 3.71 | 1.81–7.58 | 0.000 | 1.83 | 0.96–3.48 | 0.066 |
| Perceived risk | 0.07 | 0.03–0.18 | 0.000 | 0.17 | 0.05–0.54 | 0.003 |
| Trust | 4.88 | 2.67–8.90 | 0.000 | 2.82 | 1.41–5.67 | 0.004 |

OR: Odds Ratio

## Perceived usefulness of digital payments

PU was not found to be a predictor of the adoptability of digital payments among CHWs in Wakiso. Qualitative findings, however, showed that more than half of the CHWs found the mobile money payment system useful, as discussed below.

**Practical benefits and efficiency.** A significant number of the CHWs found the Mobile Money payment system to have several practical benefits, noting that it allowed them to send money virtually which enhances convenience, and the capability to manage finances more effectively by using money sparingly unlike when they had cash readily available. They also reported that digital payments save time and money required to go and pick up payments from a central point, which highlighted efficiency and practicality in the daily operations of CHWs.

> *"Digital payments systems are very useful, I can make transactions virtually, can send money easily to other people (and) they save time. I only spend on what I need (and) access to money is easier."*

(IDI, Female)

> *". . . when you use mobile money, a lot of expenses are reduced for instance when told to go and get your money from Wakiso (district headquarters), you have to (spend) transport to and fro, by the time you get the money and deduct transport, you get less than what you expected. Yet, even if I'm at a distance, I get what I expect if they use mobile money, and it also saves time."*

(IDI, Male)

**Table 5. Summary of qualitative findings on the different constructs (themes) and the sub-themes generated.**

| Themes | sub- themes |
|---|---|
| Perceived usefulness of digital payments | Practical benefits and efficiency |
| | Reliability and accuracy of payment systems |
| Perceived ease of use (PEU) of digital payments | Simplicity and support from users |
| | Shorter and streamlined procedures |
| Perceived risks (PR) associated with digital payments | Incidence of fraud and security concerns |
| | Risk of Double Payments and Financial Losses |
| Experiences of community health workers | Timeliness of digital payments |
| | Effectiveness and completeness of digital payments |
| | Trust related to digital payments |

**Reliability and accuracy of payment systems.** The payment providers reported that digital payments were useful, but they noted that with bank transfers, payments bounced less frequently than with mobile money payments since, the identification information provided for the bank accounts was more accurate and specific. On the other hand, with Mobile money payment, the CHW may provide incorrect information of a contact person against whose details they have undersigned.

*"Bank transfers are more reliable than mobile money payments due to accurate identification information, reducing payment failures and complications."*

*(KII, Payment provider)*

A few however, noted that they preferred cash payments, because unlike mobile money, they did not need to pay withdrawal fees. However, these respondents did not consider picking the cash themselves from a central point but rather the cash being delivered to them. One participant narrates,

*"Mobile money payments are easier for organizations, but as a person, I would prefer to have someone deliver the money physically: Yes, I prefer cash because I don't have to pay withdraw charges."*

*(IDI Female)*

## Perceived ease of use of digital payments

Regression results showed that PEU of digital payment systems was not a significant predictor of adoptability. Qualitative findings however revealed various opinions of whether the systems are easy to use or not, as discussed below.

**Simplicity and support for users.** Qualitative findings indicated that the majority of the CHWs found digital payment methods easy to use and learn. However, it was also reported that older people may find it hard to learn how to navigate digital systems, and would need the help of the agents or family members.

*"No, it's very simple to use. It just needs you to know your pin. Some of the agents can help you when you don't know. For example, [in the case of] Airtel where you initiate, they give you the code, . . . So, if you aren't aware of such, the agent can help you."*

*(IDI, Female)*

**Shorter and streamlined procedures.** The organizational payment providers similarly affirmed that digital payment methods were easier to learn and easier to use because of the shorter and simplified procedures and channels undertaken before the recipient receives their payment.

*"E-cash is faster. you can [spend] only one hour to pay 1000 VHTs. But that one [cash], looking for VHTs in different places, getting them is difficult because they aren't located in a central place so you have to look for them (and spend more). E-cash is better."*

*(KII-2)*

And when it comes to ease, which one is easier? Cash or this e-cash?
E-cash is easier" (KII-2)

## Perceived risks associated with the use of digital payments

PR was found to have a strong negative influence on the adoptability of digital payments. While most CHW expressed limited risks associated with digital payments, some keenly alluded to potential risks as indicated below.

**Incidence of fraud and security concerns.** Some CHWs highlighted significant concerns regarding the potential for fraud and security breaches with digital payments. One community health worker noted that they had been conned through the mobile money payment system. However, CHWs who found no risk in using digital payments noted that the risk of losing money through mobile money system usually lies with the recipient.

> *"I think that if I use my phone alone, it is upon me to take care of my phone and not leave it just anywhere . . ., even though I'm at a mobile money agent and I'm asked to input my pin. I think it's entirely my responsibility to protect my phone and my personal information."*

> *(IDI, Female)*

Some CHWs reported that they feared using cash payment systems, because of the risk of the money being stolen.

**Risk of double payments and financial losses.** The payment providers all reported that digital payments involved the risk of making double payments and a few financial losses. In most cases, they noted, these did not happen since, payments go through a verification process and double payments could be recovered during another round of payment, especially with bank payments. This recovery, however, depended on the keenness of the payment processes of the organization. Unlike digital payments, the payment providers noted that cash payments were riskier due to the higher possibilities of fraud and robbery when an individual handles cash.

> *"But also, that one [cash] I think is the [riskiest]. Because [when] you withdraw it, you may have a miscounting. . ., sometimes your objectivity gets clouded. But here as long as . . . I don't know you; I don't care who you are, I just start doing my data entry. But with cash [there is] too much power being given to the person. . . distributing the money and it's even risky for them when people know you have money. They wait for you and [rob] you."*

> *(KII-1)*

## Experiences of community health workers with digital payments

**Timeliness of digital payments.** Overall, the majority of the CHWs expressed that although they had previously received digital payments from other organizations, they experienced major delays. Some eventually receive payments when they had forgotten about the health activity for which they were receiving payment.

> *"From my experience, this digital system is far better than using the cash-based system."*

> *(IDI, Male)*

> *"What I have seen is that, apart from the delayed payments, it has been so effective; things have been normal though the only problem here is that when they say that money will be paid via mobile money, it always delays, but it comes in the end."*

> *(IDI, Male)*

*"As per those organizations which pay using mobile money, some of them I'm okay with. But some delay with the payment, . . . for example this year in September we did [the] CAST TB program but up to now we haven't been paid. We don't know why."*

*(IDI, Male)*

**Effectiveness and completeness of digital payments.** A good number of participants reported that some organizations have failed to pay the CHWs for activities done. The CHWs also reported that some organizations do not include withdrawal charges on their payments, so what is received after withdrawal is less than what was signed for. These experiences, varied from organization to organization.

*"Mobile money should be good and even all of us would love to be paid using mobile money, but it has a big challenge that has brought problems. . . Someone will bring a payment sheet and tell you to sign; that they are going to send money on the phone. After signing they don't send the money. Lawyers told us that since we signed, we received the money."*

*(IDI, Female)*

Most CHWs, however, mentioned that digital payments were better than cash payments, since they received their payments eventually and there would be no unexplained deductions.

*"[the reason] why I like the mobile money way, [is that] they will count my exact money and if I am at a workshop, they tell me that, 'You are going to get this amount.' But with cash, when it goes through so many hands, they don't tell you how much [you will get] and they (work supervisors) say 'we are the ones who will pay.' You might never know what you are supposed to get. . . ."*

*(IDI, Female)*

**Trust related to digital payments.** While most of the CHWs reported that they had trust in payment providers across common TSPs in Uganda, they noted that they didn't trust some mobile money agents/operators since they had heard of cases of people being conned and money being taken from their MM accounts while conducting transactions with these MM operators. A respondent also mentioned trusting one network more than another.

*"Yes, they [MM methods] have issues too and so many people are being [robbed] using mobile money services . . . and because of being naïve, others fall for the trap."*

*(IDI, Female)*

*"I do trust them 90%. You never know, as I told you earlier some of the agents you never know. Anything can happen any time."*

*(IDI, Female)*

The Organizational CHW payment coordinators also reported having a good experience with digital payment methods. The payment providers found digital payments faster to process than cash payments, with a greater preference for bank transfers over E-cash (MM) payment modalities. Most of them noted that they were not making any cash payments to CHWs by the

time of the interview. Concerning cash payments, the payment providers reported that most CHWs preferred MM while some CHW supervisors did not because it did not offer them the opportunity to pass some of the payments to other relatives and friends other than the intended CHWs. They also reported that MM payments are sometimes delayed due to the verification processes of the organization especially when some payments bounced.

*"I think bank transfers are faster and they are safer... you know that it is this person [the registered CHW]. If you put another name, then it is going to bounce. So, we are sure that we are paying the right person."*

*(KII-3)*

Trust was found to have a strong positive influence on the adoptability of digital payments.

## Discussion

This study found that CHWs intend to use digital payment modalities influenced by the perceived risks and trust of these modalities. CHWs reported having positive experiences with digital payment modalities but they especially complained of the delayed mobile money payments. CHW payment coordinators were similarly found to be in favor of digital payments with greater preference for bank payments.

Only Trust and Perceived Risk were found to determine the adoptability of digital payments. The dependence of trust on the subjective sense of financial security/ risk reported by Pavlou [33] appears here since CHWs developed a mistrust towards mobile money operators arising from the risk of fraud. It is worth noting however that this mistrust primarily pertains to peripheral actors in the system such as mobile money agents, rather than the central players such as payment providers or service providers. Consequently, it isn't significant enough to cause a complete refusal or lack of intention of CHWs to use digital modalities progressively as findings on the CHWs' intention to use suggest. Findings from this study are corroborated by those from another study conducted in 2020 which reported that post-use trust influences the intentions of potential users to continue to use the technology [36].

This study found a strong relationship between Perceived Risk and adoptability. The negative influence of risk on the adoptability of digital payments is similar to findings in other studies where perceived risk is reported to be a crucial factor in determining adoption of a digital system of pay [19, 37] by both the CHWs and the payment providers. Perceptions that a system or technology could be associated with adverse consequences were found to be a more powerful determinant of its adoption than the Perceived Usefulness or Perceived Ease of Use in literature. Most of the risks reported by the CHWs and payment providers were mainly related to environmental uncertainties that Pavlou [33] reports in his study on the acceptance of electronic commerce. This would imply that a lower Perceived Risk associated with digital payments would consequently lead to a greater intention to use the digital payment system overall. Despite the delays with mobile money payments, CHWs were certain that they would receive their payments. The risk of being conned was found to be a significant deterrent to digital payments but considered largely avoidable since it pertains to the individual CHW's responsibility in financial matters. Considering that Uganda is transitioning to the deployment of CHEWs (Community Health Extension Workers) in addition to VHTs, the prospects for adoption of digital payments may be even more promising given that higher levels of education are advantageous to the usability of this payment modality [5].

While the CHWs acknowledged the inconvenience caused by delays in receiving mobile money payments, the frequent experience of deductions on cash payments as they were

handed down makes CHWs hesitant to use this payment modality. In this case, cash payments are ineffective since they imply that the CHWs are paid less than they ought to. This suggests, therefore, that payment modalities that do not compromise the amount of money to be paid are preferred to those that do.

In other studies [19, 23, 29], usefulness plays a significant role as a prior determinant of adoptability. Usefulness of digital payments was explained by the convenience it provides in sending money and utilizing it. Other authors have also reported that digital payment modalities are convenient and ease the payment of online transactions [38]. However, usefulness did not have a strong influence on the adoption of digital payment systems among CHWs in Wakiso. This could perhaps be attributed to the fact that mobile money is already being used widely in other sectors, hence this is no longer a strong determinant of adoptability. Similarly, ease of use was not a strong determinant. This could also be because wide spread use of mobile money has given the CHWs prior opportunity to learn how to use the mobile payment systems. Secondly, according to the qualitative findings, the mobile money processes for accessing funds paid digitally were generally considered simple.

The adoption of digital payments offers the opportunity to leverage several benefits such as security of funds, efficiency of transfer, and better access to money by the CHWs while in field, since they are the main mobilizers of healthcare programs in communities, especially vaccination campaigns. The CHWs are recognized to be significant players in the WHO Polio Eradication Strategy [7], therefore addressing the underlying causes of mistrust and risk associated with digital payment modalities could contribute greatly to the realization of the Global Polio Eradication [7] and other community health campaigns in similar settings globally. Uganda's health system stands to benefit greatly from forging solutions to the successful application of digital payments for all health campaigns as this would lead to increased CHW motivation, health program effectiveness and accountability to stakeholders.

## Conclusion

CHWs in Wakiso district intend to use digital payment modalities especially mobile money/E-cash, therefore they can be utilized progressively for paying CHWs for community health services. Trust and perceived risk influence the intention to use digital payments among CHWs in Wakiso district. Synergized efforts by both payment providers and service operators such as TSPs to manage key identified risks among mobile money operators could attenuate perceived risk and further build trust in digital payment modalities. Despite the payment delays, CHWs appreciate the convenience provided by MM Payments in easing access to money and allowing for money spending control. Providers are keen on using digital payment modalities because they present less risk of fraud, involve shorter processes and grant greater certainty of the intended recipient receiving their pay.

### Limitations and future study

This study has limitations as it mainly focused on the relationship between independent and dependent constructs. Future studies can be conducted by using structural equation modelling to analyze the relationship between the contextual factors and the key constructs in the model. This study was conducted among CHWs in urban/semi-urban areas. Further study on similar constructs assessed in our study may be done among CHWs in more rural settings who may have less access to Digital payment services like mobile money and may rely more on cash for various transactions.

## Supporting information

**S1 Data.**
(CSV)

**S1 File. Data collection tools.**
(ZIP)

## Acknowledgments

We acknowledge the contribution of various individuals to data collection in the study: DT., HN., FLN. All authors read and approved the final manuscript.

## Author Contributions

**Conceptualization:** Veronica Kembabazi, Arnold Tigaiza, Andrew K. Tusubira.

**Data curation:** Veronica Kembabazi, Arnold Tigaiza, Andrew K. Tusubira.

**Formal analysis:** Veronica Kembabazi, Mary Nakafeero.

**Funding acquisition:** Veronica Kembabazi, Arnold Tigaiza.

**Investigation:** Veronica Kembabazi, Arnold Tigaiza.

**Methodology:** Veronica Kembabazi, Arnold Tigaiza, Charles Opio, Fredrick Edward Makumbi, Elizabeth Ekirapa Kiracho, Andrew K. Tusubira.

**Project administration:** Veronica Kembabazi, Arnold Tigaiza.

**Resources:** Veronica Kembabazi, Elizabeth Ekirapa Kiracho, Peter Waiswa.

**Supervision:** Charles Opio, Juliet Aweko, Elizabeth Ekirapa Kiracho, Andrew K. Tusubira, Peter Waiswa.

**Validation:** Veronica Kembabazi.

**Visualization:** Veronica Kembabazi.

**Writing – original draft:** Veronica Kembabazi, Arnold Tigaiza, Charles Opio, Andrew K. Tusubira.

**Writing – review & editing:** Veronica Kembabazi, Arnold Tigaiza, Juliet Aweko, Fredrick Edward Makumbi, Michael Ediau, Elizabeth Ekirapa Kiracho, Andrew K. Tusubira, Peter Waiswa.

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
