## [Decision Letter · Decision Letter 0]

27 May 2024

PONE-D-24-09580Adoptability of digital payments for Community Health Workers in peri-urban Uganda: a case study of Wakiso districtPLOS ONE

Dear Dr. Kembabazi,

Thank you for submitting your manuscript to PLOS ONE. After careful consideration, we feel that it has merit but does not fully meet PLOS ONE’s publication criteria as it currently stands. Therefore, we invite you to submit a revised version of the manuscript that addresses the points raised during the review process.

We look forward to receiving your revised manuscript.

Kind regards,

Edison Arwanire Mworozi, M.D

Academic Editor

PLOS ONE

Journal Requirements:

"Financial support for this study was obtained through the Digital Health Payments Initiative and Research project (DHPI-R) at Makerere University funded by the Bill&Melinda Gates Foundation (grant number: INV-03047). "

"Financial support for this study was obtained through the Digital Health Payments Initiative and Research project (DHPI-R) at Makerere University funded by the Bill&Melinda Gates Foundation. The findings and conclusions contained within are those of the authors and do not necessarily reflect the positions or policies of the Bill&Melinda Gates Foundation. The funders had no role in the study design, data collection and analysis, or preparation of the manuscript."

"Financial support for this study was obtained through the Digital Health Payments Initiative and Research project (DHPI-R) at Makerere University funded by the Bill&Melinda Gates Foundation (grant number: INV-03047)."

5. Please include a caption for figure 1.

Reviewers' comments:

Reviewer's Responses to Questions

**Comments to the Author**

1. Is the manuscript technically sound, and do the data support the conclusions?

Reviewer #1: Partly

Reviewer #2: Yes

2. Has the statistical analysis been performed appropriately and rigorously? 

Reviewer #1: Yes

Reviewer #2: Yes

3. Have the authors made all data underlying the findings in their manuscript fully available?

Reviewer #1: Yes

Reviewer #2: Yes

4. Is the manuscript presented in an intelligible fashion and written in standard English?

Reviewer #1: Yes

Reviewer #2: Yes

5. Review Comments to the Author

Reviewer #1: Review Reports

Review Comments

The research article is a very good paper which gives good and strong. The following are my comments;

a. On the scope” why adoptability? Is that feasible with the country context? Why only CHWs? Why only NGO workers? What about the internet governance rule of Uganda? Why only among those doing campaign? Why peri-urban? Why only polio eradication Programme?

b. The abstract lacks clarity

c. The background can be stronger than this to convince the reader.

d. The methods section is relatively good but

• Inclusion criteria is weak to cover both study methods

• Trustworthiness is not included

• Sub themes were not explained

• Case to variable ratio is great concern= for logistic regression

• The reason foe selecting TAM should be explained more

e. In the result and consequent sections

The detail character of the respondent is lacking E.g. age

The discussion is weak and weakly referenced

Some sections are missing E.g. Conflict of interest

f. Inconsistency and grammar

o IN the background “ to for”

o Use of PIN Vs pin

Regards,

Reviewer #2: ### Review Comments to the Author

**General Comments:**

- The study presents valuable insights into the adoptability of digital payments among Community Health Workers (CHWs) in Wakiso district, Uganda.

- The methodology appears sound, employing a mixed-methods approach to gather both quantitative and qualitative data, thereby providing a comprehensive understanding of the subject matter.

- The findings highlight the significant role of perceived risk and trust in influencing the adoptability of digital payment modalities among CHWs.

- The conclusion provides actionable recommendations for stakeholders to enhance the adoption of digital payment methods in the healthcare sector.

**Specific Comments:**

1. **Clarity and Organization:**

- The abstract succinctly summarizes the study objectives, methods, key findings, and conclusions.

- The flow of information is logical and well-organized, guiding the reader through the background, methods, results, and conclusion seamlessly.

2. **Methodological Rigor:**

- The use of the Technology Acceptance Model (TAM) framework adds rigor to the study by providing a theoretical basis for assessing the intention to use digital payment modalities.

- Factor analysis and logistic regression strengthen the quantitative analysis, allowing for the identification of key determinants influencing adoptability.

- Including qualitative data from Digital Payment Coordinators enriches the study by offering insights from stakeholders involved in facilitating digital payments.

3. **Key Findings:**

- The finding that nearly all CHWs have previous experience with mobile money is noteworthy and underscores the familiarity of digital payment methods among the study population.

- The identification of perceived risk and trust as critical factors influencing adoptability aligns with existing literature on technology acceptance and user behavior.

- The qualitative findings regarding positive experiences with mobile money, despite some delays, provide valuable context for understanding CHWs' attitudes toward digital payments.

4. **Recommendations:**

- The recommendations for collaborative efforts between payment providers and service operators to address payment delays and mitigate risks are actionable and align with the identified barriers to adoptability.

- Suggesting strategies to enhance trust and alleviate perceived risks, such as transparent communication and risk management protocols, demonstrates a practical approach to promoting digital payment adoption.

**Overall Impression:**

- The study makes a significant contribution to the literature on digital payment adoption in the healthcare sector, particularly among CHWs in peri-urban settings.

- The findings are relevant not only to stakeholders in Wakiso district but also to policymakers and practitioners seeking to leverage digital solutions to improve healthcare delivery and financial inclusion.

- Addressing the review comments and incorporating any necessary revisions will enhance the clarity, rigor, and impact of the study.

6. PLOS authors have the option to publish the peer review history of their article (what does this mean?). If published, this will include your full peer review and any attached files.

Reviewer #1: No

Reviewer #2: No

---

## [Author Response · Author response to Decision Letter 0]

19 Jul 2024

Reviewer #1: Review Reports

1. The research article is a very good paper which gives good and strong. 

Response: Thank you

2. On the scope” why adoptability? Is that feasible with the country context? Why only CHWs? Why only NGO workers? What about the internet governance rule of Uganda? Why only among those doing campaign? Why peri-urban? Why only polio eradication Programme?

Responses: 

“why adoptability?”: We chose to assess adoptability because the CHWs have used digital payments in previous community health activities and campaigns as well as in their day to day lives. We, therefore, wanted to see if they are willing to continue using digital payments for future community health projects. This is indicated in the last paragraph of the introduction section, line 105-110. 

“Is that feasible with the country context?”: Yes, adoptability of digital payments is feasible in Uganda. There are various forms of digital payments in Uganda, however, Mobile Money transfers run by telecommunication companies are the most feasible and readily available. These only requires one to have a phone (smart phone or not) and they can receive and send their money. We have connectivity of telecommunication services including mobile money agents across the country. Banks are also available across the country though not all the CHWs have accounts. 

“Why only CHWs?” Salaried cadres of the healthcare system in Uganda already receive payments digitally through bank transfers. CHWs, however, who are engaged and paid occasionally, often receive their money as cash. Payment with cash has been associated with several challenges as indicated in the introduction (line 101-104), such as time wasting as the wait for money at facilities since they’re always in large numbers, transportation of cash from the funders to the distribution centers is often a problem and sometimes results in loss of some funds, the accountability is a problem, and the money is usually not received in full amounts. Digital payments will address these problems in this population. In addition, CHWs are the lowest cadre in the health system, with lower literacy levels so, use of banks like for other cadres could be a challenge. 

“Why only NGO workers?” The study focused on CHWs who had ever been paid digitally and usually, payments are given by NGOs through their various community projects. Therefore, our inclusion criteria included having received payments digitally and the listed NGOs were some of the biggest funders of recent and ongoing community health projects. UNEPI coordinates government immunization programs in which CHWs are often engaged. 

“What about the internet governance rule of Uganda?” Digital payments systems mostly used in Uganda as well as the mobile money systems being recommended in the current study do not require internet. The issue of vulnerability to internet hacks and scams is therefore not significant. Besides that, CHWs as well as majority of the population in Uganda already use digital payments in their day to day lives. We wanted to see if these same services are adoptable in the health sector for payment of CHWs for their involvement in health activities. 

“Why only among those doing campaign?” CHWs are basically volunteers. Majority of their paid activities are community health campaigns such as immunization/vaccination. Therefore, at the time of sampling, majority or all of the CHWs who had received digital payments had been engaged in campaign activities (Line 67-73).

“Why peri-urban?” Peri-urban because it gives a good representation of both the urban and the not so urban setting as a proxy for rural settings to show the adoptability in both settings. The findings on adoptability in peri-urban areas gives a starting point for the initiation of adoption of digital payments and scale up to other rural areas as guided by phone coverage. 

“Why only polio eradication Programme?” These were emphasized by WHO when recommending digital payments however these payment modalities can be used in other community health campaigns. Digital payments apply to other health campaigns besides polio eradication programmes. 

3. The abstract lacks clarity

Response: Thank you. The abstract was edited to provide more clarity.

4. The background can be stronger than this to convince the reader.

Response: Thank you, we have made some additions to the background to make it stronger as guided. 

5. The methods section is relatively good but;

• Inclusion criteria is weak to cover both study methods

Response: Thank you, this has been addressed. This survey was conducted among CHWs in Wakiso district who were carrying out community health activities in the district supported by various Implementing Organizations while In-depth interviewees were CHWs with varied experiences with digital payments and willing to share and Key informant interviewees were coordinators of CHWs and individuals who played key roles in management of digital payments for CHWs such as payment providers. See lines 135 – 138.

6. Trustworthiness is not included

Response: Thank you. Trustworthiness is captured under “Trust” which is a main construct in the conceptual model for this study. Trust is explained in the methods, under the variables section as well as in the analysis plan and the results. 

7. Sub themes were not explained

Response: The sub-themes were added to the results section under each of the constructs (themes). See lines 294-368. 

8. Case to variable ratio is great concern= for logistic regression

Response: "The final model had 4 variables and a total of 150 observations which gave an average of 38 cases per variable. This satisfies the working assumption of 10-20 cases per variable in the model. (Ja, 2017)

9. The reason for selecting TAM should be explained more

Response: The reasons for selecting TAM were explained in the study design between line 145 to 153. We have also added that the TAM could fit the local context. 

10. In the result and consequent sections, the detail character of the respondent is lacking E.g. age

Response: Thank you, details on age and gender have been added in the results section. See Table 3.

11. The discussion is weak and weakly referenced

Response: Thank you, the comment has been addressed. 

12. Some sections are missing E.g. Conflict of interest

Response: Conflict of interest statement “The authors have declared that no competing interests exist,” was made in the submission portal as guided but the system. The PLOS One formatting guidelines indicate this, “Do not include funding or competing interests information in Acknowledgments” and there seems not to be provision for a section specifically for Conflict of interest in the same guidelines.

13. Inconsistency and grammar

Response: Thank you, the manuscript has been reviewed by a professional language editor.

14. IN the background “ to for”

Response: Thank you, this has been edited.

15. Use of PIN Vs pin

Response: Thank you, this has been edited.

Reviewer #2: ### Review Comments to the Author

1. The study presents valuable insights into the adoptability of digital payments among Community Health Workers (CHWs) in Wakiso district, Uganda. - The methodology appears sound, employing a mixed-methods approach to gather both quantitative and qualitative data, thereby providing a comprehensive understanding of the subject matter. - The findings highlight the significant role of perceived risk and trust in influencing the adoptability of digital payment modalities among CHWs. - The conclusion provides actionable recommendations for stakeholders to enhance the adoption of digital payment methods in the healthcare sector.

Response: Thank you for appreciating

2. Clarity and Organization: - The abstract succinctly summarizes the study objectives, methods, key findings, and conclusions. - The flow of information is logical and well-organized, guiding the reader through the background, methods, results, and conclusion seamlessly.

Response: Thank you for observing this

3. Methodological Rigor: ** - The use of the Technology Acceptance Model (TAM) framework adds rigor to the study by providing a theoretical basis for assessing the intention to use digital payment modalities. - Factor analysis and logistic regression strengthen the quantitative analysis, allowing for the identification of key determinants influencing adoptability. - Including qualitative data from Digital Payment Coordinators enriches the study by offering insights from stakeholders involved in facilitating digital payments.

Response: Thank you for noticing this

4. Key Findings: ** - The finding that nearly all CHWs have previous experience with mobile money is noteworthy and underscores the familiarity of digital payment methods among the study population. - The identification of perceived risk and trust as critical factors influencing adoptability aligns with existing literature on technology acceptance and user behavior.

- The qualitative findings regarding positive experiences with mobile money, despite some delays, provide valuable context for understanding CHWs' attitudes toward digital payments.

Response: Thank you for appreciating

5. **Recommendations: ** - The recommendations for collaborative efforts between payment providers and service operators to address payment delays and mitigate risks are actionable and align with the identified barriers to adoptability. - Suggesting strategies to enhance trust and alleviate perceived risks, such as transparent communication and risk management protocols, demonstrates a practical approach to promoting digital payment adoption.

Response: Thank you for noticing this.

6. **Overall Impression: ** - The study makes a significant contribution to the literature on digital payment adoption in the healthcare sector, particularly among CHWs in peri-urban settings. - The findings are relevant not only to stakeholders in Wakiso district but also to policymakers and practitioners seeking to leverage digital solutions to improve healthcare delivery and financial inclusion. - Addressing the review comments and incorporating any necessary revisions will enhance the clarity, rigor, and impact of the study.

Response: Thank you for the observations

References

Ja, R. (2017). How many independent variables to include BEFORE running logistic regression? Retrieved from https://www.researchgate.net/post/How-many-independent-variables-to-include-BEFORE-running-logistic-regression/59bbc2ab96b7e4d751391ff0/citation/download

---

## [Editor Report · Decision Letter 1]

23 Jul 2024

Adoptability of digital payments for Community Health Workers in peri-urban Uganda: a case study of Wakiso district

PONE-D-24-09580R1

Dear Kembabazi

We’re pleased to inform you that your manuscript has been judged scientifically suitable for publication and will be formally accepted for publication once it meets all outstanding technical requirements.

Kind regards,

Edison Arwanire Mworozi, M.D

Academic Editor

PLOS ONE
---

## [Editor Report · Acceptance letter]

6 Aug 2024

PONE-D-24-09580R1 

PLOS ONE

Dear Dr. Kembabazi, 

I'm pleased to inform you that your manuscript has been deemed suitable for publication in PLOS ONE. Congratulations! Your manuscript is now being handed over to our production team.

Kind regards, 

on behalf of

Professor Edison Arwanire Mworozi 

Academic Editor

PLOS ONE